# Fifty Years of Animal Toxin Research at the Shemyakin–Ovchinnikov Institute of Bioorganic Chemistry RAS

**DOI:** 10.3390/ijms241813884

**Published:** 2023-09-09

**Authors:** Victor Tsetlin, Irina Shelukhina, Sergey Kozlov, Igor Kasheverov

**Affiliations:** 1Department of Molecular Neuroimmune Signaling, Shemyakin-Ovchinnikov Institute of Bioorganic Chemistry, Russian Academy of Sciences, 16/10 Miklukho-Maklay Str., 117997 Moscow, Russia; shelukhina.iv@yandex.ru (I.S.); shak_ever@yahoo.com (I.K.); 2Department of Molecular Neurobiology, Shemyakin-Ovchinnikov Institute of Bioorganic Chemistry, Russian Academy of Sciences, 16/10 Miklukho-Maklay Str., 117997 Moscow, Russia; serg@ibch.ru

**Keywords:** animal toxins, snake, scorpion, spider, ion channel, receptor

## Abstract

This review covers briefly the work carried out at our institute (IBCh), in many cases in collaboration with other Russian and foreign laboratories, for the last 50 years. It discusses the discoveries and studies of various animal toxins, including protein and peptide neurotoxins acting on the nicotinic acetylcholine receptors (nAChRs) and on other ion channels. Among the achievements are the determination of the primary structures of the α-bungarotoxin-like three-finger toxins (TFTs), covalently bound dimeric TFTs, glycosylated cytotoxin, inhibitory cystine knot toxins (ICK), modular ICKs, and such giant molecules as latrotoxins and peptide neurotoxins from the snake, as well as from other animal venoms. For a number of toxins, spatial structures were determined, mostly by ^1^H-NMR spectroscopy. Using this method in combination with molecular modeling, the molecular mechanisms of the interactions of several toxins with lipid membranes were established. In more detail are presented the results of recent years, among which are the discovery of α-bungarotoxin analogs distinguishing the two binding sites in the muscle-type nAChR, long-chain α-neurotoxins interacting with α9α10 nAChRs and with GABA-A receptors, and the strong antiviral effects of dimeric phospholipases A2. A summary of the toxins obtained from arthropod venoms includes only highly cited works describing the molecules’ success story, which is associated with IBCh. In marine animals, versatile toxins in terms of structure and molecular targets were discovered, and careful work on α-conotoxins differing in specificity for individual nAChR subtypes gave information about their binding sites.

## 1. Introduction

Animal venoms, over a long period of evolution, have become a powerful means of attack and defense. They consist mostly of proteins and peptides, usually called toxins. The best-known terrestrial venomous animals are snakes, scorpions, and spiders. Some invertebrate marine animals like *Conus* snails, jellyfishes, and sea anemones are also very successful in toxin production. For many centuries, the deadly bites and stings of these animals have been attracting the attention of naturalists and natural medicine specialists. However, it was only in the last century that the study of venoms at the molecular level began. The individual toxins were isolated from venoms and their structures and functions were investigated. Due to their high affinity and selectivity towards various receptor targets, animal toxins are widely used as versatile tools for the study of different physiological processes at the molecular level. Among many others, especially potent was α-bungarotoxin (αBgt), a neurotoxin isolated 60 years ago from the venom of krait *Bungarus multicinctus* [1], which was used as a first tool in the studies of nicotinic acetylcholine receptor (nAChR) [2]. In the current jubilee Special Issue, we shall consider the research on toxins and receptors performed at the Shemyakin–Ovchinnikov Institute of bioorganic chemistry of the Russian Academy of sciences (IBCh) in Moscow, which started approximately 10 years after the discovery of αBgt—which explains the title of our review. We will discuss various proteins from natural venoms, including those that, like αBgt, share a three-finger spatial structure and interact with nAChRs. The studies of protein and peptide toxins from venomous animals carried out at IBCh will be accompanied by those of their synthetic analogs.

## 2. Structural Studies of Animal Toxins

### 2.1. Primary Structure Determination

The first results of the work on the protein toxins at IBCh were the determination of the primary structures of several toxins from the Middle Asian cobra *Naja oxiana*. Among them were α-neurotoxins, namely short-type neurotoxin II and long-type neurotoxin I [3,4], as well as cytotoxins [5,6] (Figure 1). It should be noted that, in those days, there was no such reliable assistant as modern mass-spectrometry to check the purity and to determine the structure of the obtained peptide or protein: the isolation of an individual toxin required a combination of various chromatography techniques, while Edman degradation was in fact the sole method for primary structure determination. The use of mass-spectrometry greatly facilitated this task and, for example, allowed the detection of the first tryptophan-containing “weak” three-finger toxin (TFT) WTX in *Naja kaouthia* venom [7] (Figure 1). Unusual forms of snake venom toxins were found at our institute considerably later, and among them were muscarinic toxin-like proteins [8] (Figure 1) and glycosylated cytotoxin [9], the first representative of glycosylated TFTs. All the above-mentioned toxins are from the TFT family. However, snake toxins from other families were discovered as well. They included heterodimeric neurotoxic phospholipases A2 (PLA2) from the venom of viper *Vipera nikolskii* [10], as well as azemiopsin, a peptide neurotoxin of the new type [11], and original bradykinin-potentiating peptides from viper *Azemiops feae* [12] (Figure 1). 

At the same time, intensive experiments were carried out to study toxic molecules in arthropod venoms. Due to the limitations of the biological tests available, the search for active molecules has mainly been based on toxicity to mammals and/or insect species. Fifteen different toxins from *Buthus eupeus* scorpion venom, among them insectotoxins I1, I3, I4, and I5, belonging to yet unknown structural type [13], and three polypeptides, M10, M14, and M9, toxic to mammals, were isolated [14] (Figure 1). The complete amino acid sequence of the 66-amino-acid-residue toxins M9 and M14 was established by detecting overlapping peptide fragments using Edman degradation [14]. Another four polypeptide neurotoxins, possessing paralytic activity towards mice, were isolated from the *Orthochirus scrobiculosus* scorpion’s venom, and, for the major compound Os-3, after tryptic and chymotryptic cleavage, the complete amino acid sequence was established [15].

The development of molecular biology methods (cloning and sequencing of DNA) and mass-spectrometry made protein sequence determination more technologically advanced and productive. With these approaches, many more toxin sequences were determined, including those of phospholipases A2 from snake venoms [10,16], as well as other classes of toxins from scorpion and spider venoms. In particular, the isolation of the toxin named OsI-1, with a molecular weight of 6994 Da, responsible for the *O. scrobiculosus* scorpion venom’s toxicity to insects, and its partial N-terminal sequence have served as the starting point for gene cloning. The complete primary structure of OsI-1 (Figure 1) was deduced from the cDNA sequence obtained by the rapid amplification of cDNA ends (RACE) method. The mass spectroscopy data indicated the post-translational modification of the precursor protein at maturation by three C-terminal amino acids’ truncation and the amidation of the C-terminus [17]. 

Other examples are shorter polypeptide toxins BeKm-1 and OSK1 from scorpion venoms (Figure 1); using the same approach, their structures were derived from the nucleotide sequence of the mRNA encoding the complete precursor protein. Further, the examination of BeKm-1’s effects on different Kv channels revealed its selective inhibitory effect on hERG1 ion channels with an IC_50_ of 3.3 nM, which made it the first selective inhibitor of this channel [18]. In contrast, the OSK1 inhibitory activity had a lack of selectivity: determined initially as small-conductance Ca-activated K^+^-channels inhibitor [19], this toxin blocked the conductance of voltage-gated potassium channels, the intermediate conductance of calcium-activated potassium channels, and even the mouse muscle nAChR in the micromolar concentration range [20].

This approach also made it possible to determine the complete primary structures of several new toxic compounds found in the venom of sea anemones. Among the most recent was the identification of a series of peptides (Ms11a-1/4) from *Metridium senile* capable of inhibiting nAChR (Figure 1), three of which demonstrated sub-micromolar affinity for the muscle-type receptor, and one (Ms11a-3) showed maximum efficacy of interaction with α9α10 nAChR [21].

The transition of primary structure determination for toxins from direct sequencing by Edman degradation to mRNA data decoding revealed the heterogeneity of the mRNA in arachnids long before the discovery of gene multiplicity in their genomes. The study of the structural variability in the Os3 neurotoxin from *O. scrobiculosus* revealed the family of cDNAs that encode the highly homologous Os3-like polypeptides [22]. The further development of techniques and cost reductions for nucleic acid sequencing made it possible to analyze the mRNAs’ multiplicity in the venom glands and the creation of the so-called expressed sequence tag (EST) library. 

A pipeline for the processing of large data collected from EST was essential in obtaining the structural information. In IBCh, such an approach for the primary structure of mature toxin search was suggested for spiders’ toxins and further expanded to other venomous species [23,24]. Thus, a comprehensive transcriptome analysis of the EST database was performed for spiders *Agelena orientalis* and *Dolomedes fimbriatus* [23,25]. For *A. orientalis*, by the proteomic analysis of venoms milked individually from a single spider, a lower number of polypeptides was found in each sample than was expected from the transcriptome analysis. However, it was found that the composition of the venom obtained by summing the proteomic data for 20 individual animals was similar in composition to that from the transcriptomics data, but each spider individually produced a limited pool of toxins from the general assortment [26]. 

Special mention can be made of the discovery of the largest animal toxins named latrotoxins. Dangerous to humans, the venom of the black widow spider *Latrodectus mactans tredecimguttatus* contains high-molecular-weight toxins (above 150 kDa). The construction of a cDNA library for spider venom glands led to the determination of the primary structure of the 1401-amino-acid α-latrotoxin precursor, from which the first toxin sequence was derived [27]. Further, α-latroinsectotoxin, a presynaptic neurotoxin selective only for insects and composed of 1411 amino acids, was sequenced. It was shown that the mature α-latroinsectotoxin with a molecular mass of approximately 130 kDa was produced by processing in the N-terminal and C-terminal regions of the precursor [28]. A few years later, another insect-specific toxin, δ-latroinsectotoxin, that contained 1214 amino acid residues, was sequenced [29] (Figure 1). The domain organization of all latrotoxins was found to be similar, suggesting that the toxins are a family of related proteins. α-Latrotoxin has made an invaluable contribution to the development of knowledge about the mechanism of synaptic vesicle exocytosis in presynaptic nerve terminals. It is capable of stimulating exocytosis, via binding to two distinct families of neuronal cell surface receptors, neurexins and latrophilins, which share an important function in synaptic cell adhesion [30].

### 2.2. Toxin Spatial Structure Studies

In the early 1970s, there was no information about the spatial structure of snake venom neurotoxins. At that time, at IBCh, the laboratory of nuclear magnetic resonance was led by well-known biophysicist Prof. V. Bystrov; the dependence of the vicinal coupling constant in peptide ^1^H-NMR on the dihedral angle is known as the Karplus–Bystrov relationship. It was hoped that some information about the spatial structure of α-neurotoxins could be obtained by the ^1^H-NMR method, while the chemically modified derivatives of α-neurotoxins would assist the signal assignment in the spectra. It was established that the central loop II and the C-terminal loop III of the short α-neurotoxin NT-II from the *N. oxiana* venom were in proximity [31] (Figure 2A), but, at that time, ^1^H-NMR could not solve the total structure even in relatively small proteins. However, this conclusion was in accordance with the complete structure of erabutoxins, the short-chain α-neurotoxins, which was established at that time in the USA by X-ray analysis [32,33]. 

A positive circumstance that contributed to determining a large number of NMR structures of various TFTs at IBCh was the development in the department led by Prof. M. Kirpichnikov of efficient methods for the expression of naturally occurring neurotoxins and their mutants. Among such proteins were short- and long-chain α-neurotoxins and non-conventional toxins [34,35]. Improvements in NMR methods, as well as the possibility of obtaining recombinant toxins and their mutants, led to the determination of a number of new structures at the department led by Prof. A. Arseniev—for example, the complete structures of naturally occurring neurotoxin II and the non-conventional WTX mutant [36,37] (Figure 2A,B).

In collaboration with Dutch colleagues, the X-ray structure of an unusual dimeric form of α-cobratoxin, where two long-type TFT monomers were connected by two intermolecular disulfides, was determined [38,39] (Figure 2C). 

The development of NMR instrumentation allowed the determination of high-resolution structures of peptide and protein neurotoxins. Thus, the first 3D structure of neuronal α-conotoxin ImI in solution was determined [40], and recently the spatial structure of peptide neurotoxin apamin was confirmed [41] (see Figure 2).

For a number of peptide toxins from scorpion venoms, the spatial structures were also resolved by the ^1^H-NMR technique at IBCh. These include OSK1 from the *O. scrobiculosus* venom, mentioned above as a nonselective blocker of small-conductance Ca-activated K^+^ channels [42] (Figure 2D); mammalian neurotoxin M9 from the *B. eupeus* venom, which was shown to have two slowly exchangeable conformations at acidic pH [43]; the insectotoxin I5A from the *B. eupeus* containing the α-helical and anti-parallel β-structure [44]; and potassium channel inhibitor BeKm-1 from *B. eupeus*, which also comprised a short α-helix and a triple-stranded antiparallel β-sheet [45]. The obtained BeKm-1 structure allowed the initiation of the design of point-mutated analogs, followed by the identification of residues important for the binding of BeKm-1 to the human ERG (HERG) potassium channel. Their unusual location in the α-helix and following loop was established, contrary to the β-sheet region in “traditional” scorpion toxins [45]. 

Several 3D structures of arachnid toxins were determined by NMR as well. Among them, the structure of a modular arachnid toxin, purotoxin-2 (PT2), from the wolf spider *Alopecosa marikovskyi* (Lycosidae) should be mentioned [46] (Figure 2F). This toxin, which comprises an N-terminal inhibitor cystine knot (ICK) β-structural domain and a C-terminal linear cationic domain, is the founding member of a large family of polypeptides with similar structural motifs.

A non-standard variant of ICK folding was detected by an NMR study of the nAChR-targeting peptides Ms11-2 and Ms11a-3, identified in *M. senile* sea anemone venom (see above). A structural feature of both compounds was the presence of a prolonged loop between the fifth and sixth cysteines [21] (Figure 2E).

## 3. Three-Finger Proteins from Snake Venoms in Research on nAChRs

### 3.1. Overview of α-Neurotoxin–nAChR Relationships

nAChR was the first ion channel receptor to be characterized in detail, and a huge role in this was played by the discovery of αBgt [1], a sophisticated tool for the detection and further detailed investigation of this receptor. Soon, it became clear that αBgt is a member of a large family of diverse protein neurotoxins present in the venoms of different snakes, which may have higher or lower selectivity to different nAChR subtypes. Originally, αBgt was used to block the activity of the muscle-type nAChR from the fish electric organs [2] and, later, radioactive and fluorescent derivatives of αBgt played a role as excellent tools in detecting and measuring the levels of the nAChRs. In particular, with the aid of radioiodinated αBgt, the presence of nAChR, later identified as neuronal α7 nAChR, was shown in the brain [47]. Chromatography on an affinity column with α-cobratoxin (belonging, similarly to αBgt, to the group of the so-called long-chain α-neurotoxins) was one of the methods of nAChR purification [48]. Current knowledge of the nAChR structure and function, including the mechanisms of the toxin–receptor interactions, is presented in textbooks and recent reviews [49,50,51]. Elucidation of the nAChR three-dimensional structure took a long way: from the nAChR low-resolution X-ray structure [52] to the high-resolution X-ray structure of the acetylcholine-binding proteins (AChBPs) [53], an excellent structural model of the ligand-binding domains not only nAChRs but of all Cys-loop ligand-gated ion channels. This important step was followed by the first refined cryo-electron microscopy structure of the *Torpedo marmorata* ray nAChR [54] and then by high-resolution X-ray structures of microbial proteins GLIC and ELIC [55,56]. The next successes were the high-resolution structures of a couple of neuronal heteromeric nAChRs (on which α-neurotoxins are not acting or acting with low affinity) in complexes with low-molecular agonists and antagonists (see, for example, [57]). Regarding α-neurotoxins, until recently, their complexes with either the AChBPs (built from five identical subunits) [58], pentameric chimeric ligand-binding domain (LBD) of the α7 nAChR subunit [59], or monomeric LBDs of the α1 [60] and α9 [61] nAChR subunits were known. Only in the last two years, cryo-electron microscopy provided the high-resolution structures of αBgt in complex with the muscle-type *Torpedo* receptor [62] and with the human neuronal α7 nAChR [63], as well as a complex of a short-chain α-neurotoxin with the *Torpedo* nAChR [64]. This briefly presented information reflects the current state of the α-neurotoxin–nAChR relationships.

### 3.2. Interaction of α-Neurotoxins Bearing Fluorescent, Spin, or Photoactivatable Labels with the nAChRs

The relatively large amounts of isolated snake venom proteins available at IBCh made possible the preparation of their chemically modified derivatives to obtain information about their spatial structures and for the analysis of the mechanisms of action. Starting from neurotoxin II, a series of singly labeled derivatives containing a fluorescent or spin label at the identified amino acid residues were obtained and used to map the topography of the neurotoxin binding to the *T. marmorata* nAChR. With the aid of fluorescence and EPR spectroscopy, several labels forming contacts with the receptor were identified, including those on Lys27 in the central loop II and on Lys47 in the C-terminal loop III [65]. In most of the mentioned X-ray and cryo-electron microscopy structures, the toxin was a long-chain α-neurotoxin (α-cobratoxin or α-bungarotoxin), and the major role in contacts with the receptor or its models was played by the toxin central loop II. 

Later, in collaboration with Prof. F. Hucho (Free University, Berlin), a series of photoactivatable derivatives of short- and long-chain α-neurotoxins was synthesized and their contacts with various nAChR subunits of the *Torpedo californica* were characterized [66,67,68]. The results obtained in these experiments also revealed the predominant participation of the toxins’ central loop II in the interaction with nAChR, and, in the case of short-chain neurotoxins, also loop III, which was fully confirmed after publishing the cryo-electron structures of complexes of the *Torpedo* receptor with both long-chain [62,63] and short-chain [64] neurotoxins.

### 3.3. Recent Studies on Snake Venom α-Neurotoxins

More recent work on snake venom neurotoxins involved the demonstration of α-cobratoxin’s capacity to block some subtypes of the ionotropic GABA-A receptors [69]. Using, in competition with α-cobratoxin, its mutant and synthetic fragments provided evidence for the involvement of the α-cobratoxin central loop II in binding to the GABA-A receptor. This conclusion is in agreement with the recently solved cryo-electron microscopy structure of the GABA-A receptor complex with α-cobratoxin [70]. The search for toxins acting both on the nAChRs and GABA-A receptor continued and several such novel proteins were isolated from the venom of cobra *Naja melanoleuca* [71]. Concerning α-bungarotoxin, whose anniversary is celebrated in the present Special Issue, a collaboration with German colleagues should be mentioned, where several new variants of α-bungarotoxin have been isolated, which, contrary to α-bungarotoxin itself, have a different affinity for the two orthosteric binding sites in the *T. californica* nAChR [72].

At present, a great role among the fluorescent labels is played by green fluorescent protein (GFP) and related proteins. At IBCh, GFP was recombinantly fused with α-cobratoxin for research on the nAChR. However, the results were unsatisfactory and a much higher resolution in the detection of α7 and muscle-type nAChRs in the cell lines was achieved when α-cobratoxin was chemically modified not with full-size GFP but only using its synthesized chromophore [73]. 

## 4. Other Snake Venom Toxins

### 4.1. Cobra Venom Cytotoxins: Spatial Structure and Biological Activity

Cytotoxins (cardiotoxins) belong to the TFT family and are found exclusively in cobra venoms. These toxins nonspecifically kill cells by attacking and destroying the cell membrane. A few cytotoxins from cobra *N. oxiana* venom were purified and sequenced at IBCh in the early seventies of the last century [5,6] (Figure 1). Further studies were directed to the determination of the detailed spatial structures of the cytotoxins and the molecular mechanisms of their biological effects. Using different variants of NMR spectroscopy, the spatial structures of cytotoxins 1 and 2 in solution and in the membrane environment were determined [74,75,76]. The differences in the mode of interaction with the membrane between different cytotoxin types were revealed and the important role of polypeptide loops I and II was recently demonstrated [77]. The first, and so far unique, among three-fingered toxins, glycosylated cytotoxin was isolated from cobra *N. kaouthia* venom, where the glycosylation of Asn29 in loop II resulted in a decrease in its biological activity [9]. It was found that the cytotoxicity depended on the structure of loops I and II, the toxins with two adjacent proline residues in loop I being the least active [78,79]. Using the confocal spectral imaging technique and fluorescently labeled cytotoxins, it was shown that cytotoxins penetrated readily into cells and accumulated markedly in the lysosomes, and this accumulation correlated well with the cytotoxic effects [79].

The biological activity of cytotoxins is quite diverse. Thus, they have been shown to manifest antiprotozoal activity [80]. The cardiotoxicity of cytotoxins is well known. However, recent works at IBCh, in collaboration with other institutes of RAS, showed that this activity is also dependent on the structural types of cardiotoxins [81,82]. These differences were observed in the whole heart [82], isolated muscle and aorta [81], as well as in cardiomyocytes [83]. It was shown that the effect of cytotoxins on the cardiovascular system was mainly associated with the disruption of the transport systems responsible for Ca^2+^ influx [81].

### 4.2. Non-Conventional Three-Finger Neurotoxin

So-called “weak” snake venom toxins were known long ago and obtained this name because they were virtually nontoxic and their targets were not known. Contrary to long-chain α-neurotoxins, these proteins have an additional disulfide not in loop II, but in loop I. One of these toxins—weak toxin (WTX)—was isolated at IBCh [7] (Figure 1 and Figure 2B), and, in collaboration with Prof. D. Bertrand (Switzerland) and Dr. C. Methfessel (Germany), for the first time, it was demonstrated that WTX at micromolar concentrations inhibited both the muscle-type and neuronal α7 nAChRs [84]. The toxins of this type are now called non-conventional toxins. Further studies demonstrated that among WTX targets can also be some subtypes of muscarinic acetylcholine receptors [37,85]. Because WTX had also effects on blood pressure [86] and was almost nontoxic, it can be considered as a basis for a potential drug. 

### 4.3. Three-Finger Mammalian Proteins

Three-finger proteins are also present in diverse organisms, including mammals, forming the so-called Ly6/uPAR family (see reviews [87,88]). They are mentioned here, close to WTX, because, structurally, they are most similar to nonconventional toxins having the additional fifth disulfide in the N-terminal loop I, but not in the central loop II, like the long-chain α-neurotoxins (see Figure 2B,C). Most of the Ly6/uPAR proteins, like Lynx1, have the glycosylphosphatidylinositol (GPI) anchor at their C-terminus, by which they attach to the membrane, but secreted proteins also exist, like SLURP1, lacking this anchor. Although these proteins do not belong to the animal toxins considered in the present review, we will present brief information about those that act on nAChRs. The reason for their inclusion is that the work at IBCh on the ^1^H-NMR spatial structures of Lynx1 and SLURP1 analogs, expressed in *E. coli,* was based on previous work both on the naturally occurring TFTs (see Section 2.2) and those expressed in *E. coli*. The experience in the production of TFTs in *E. coli* and their ^1^H-NMR analysis in the departments of Prof. A. Arseniev and Prof. M. Kirpichnikov allowed, for the first time, the establishment of the structures of SLURP-1 and wsLynx1 (water-soluble Lynx1 analog lacking the GPI tail) [89,90]. In the present review, focusing on toxins from animal venoms, we do not consider in detail the interactions of these proteins with the muscle-type and neuronal nAChRs, but we should mention that an analysis of their activity was performed using the two-electrode voltage-clamp method and by competition with radioiodinated α-bungarotoxin, which allowed to distinguish their interactions with the orthosteric and allosteric binding sites in these receptors [89,91,92].

### 4.4. Phospholipases A2

The first phospholipase A2 (PLA2) was isolated from the cobra *N. oxiana*’s venom at IBCh in 1977 [93]; later, its complete amino acid sequence was determined [94]. The determination of the amino acid sequences for PLA2s from other snake venoms was mentioned in Section 2.1 [7,8] (see Figure 1). The studies of PLA2s’ biological activity discovered new properties. It has been shown that the noncytotoxic PLA2 from the *N. haje* cobra venom is the first thrombin inhibitor of the PLA2 family [95]. PLA2s were found to inhibit nAChRs and this capability did not depend on their enzymatic activity [96,97]. Phospholipase A2 from krait *Bungarus fasciatus* venom manifested cytotoxicity against human cancer cells in vitro [98]. Recently, the antiviral activity of several snake venom PLA2s was studied. It was found that dimeric PLA2 from the viper *Vipera nikolskii* and its subunits demonstrated potent virucidal effects, which were related to their phospholipolytic activity and interfered with the binding both of an antibody against angiotensin-converting enzyme 2 (ACE2) and of the receptor-binding domain of the SARS-CoV-2 virus’ glycoprotein S to 293T/ACE2 cells [99]. Dimeric PLA2s had also pronounced virucidal and anti-HIV activity. They inhibited syncytium formation between chronically HIV-infected cells and healthy CD4-positive cells and blocked HIV binding to cells [100], this action being dependent on their catalytic activity. Thus, snake venom PLA2s might be considered as candidates for lead molecules in antiviral drug development.

### 4.5. Linear Peptides

Finishing this section dealing with snake toxins, it is worth mentioning azemiopsin—a peptide that was isolated at IBCh from the *Azemiops feae* venom [11] (Figure 1). It does not contain disulfides but quite selectively inhibits the muscle-type nAChRs [11] and, according to preclinical studies, is an efficient myorelaxant [101]. Two new bradykinin-potentiating peptides possessing unique amino acid sequences were isolated from the venom of *A. feae* [12] (see Figure 1). They have no consensus C-terminal sequence PPIPP but efficiently potentiate the effect of bradykinin.

It should be mentioned that linear toxins (lacking disulfide bonds) were also characterized in the venoms of some spider species. Particularly large content of such peptides having large structural variability was found in the venom of the *Lachesana tarabaevi* spider and these toxins were named latarcins [102]. Further studies of the spider superfamily Lycosoidea revealed other numerous linear toxins in the venoms with predominantly membrane-destroying activity and cytotoxicity to bacterial and partially to mammalian cells. The mechanisms of interaction with cellular membranes and selectivity to the composition of model lipid membranes were later established for these molecules and described in detail [103]. From the latarcin pool, latarcin 2a (see Figure 1) was the most active among others and had higher cytotoxicity to bacteria as well as to red blood cells.

## 5. Marine Toxins Acting on the Ligand- or Voltage-Gated Ion Channels

### 5.1. α-Conotoxins in Research on nAChRs

α-Conotoxins, small neurotoxic peptides with two disulfide bridges found in venoms of the predatory marine mollusks of the *Conus* genera, emerged in research on nAChRs more than 40 years ago [104,105], and their advantage is their capacity not only to distinguish muscle-type nAChRs from neuronal ones but also to be markers of individual neuronal nAChRs (see, for example, reviews [106,107,108]). Earlier work at IBCh, starting in the 1990s, on the synthesis and analysis of the spatial structures of α-conotoxins [40,109,110] (Figure 2H) was performed in collaboration with Dr. C. Methfessel (Germany). Later, using photoactivatable derivatives of α-conotoxins, their contacts with various subunits of the *T. californica* nAChR were demonstrated [111,112,113], resulting in the mapping of the orthosteric binding sites of this receptor [113]. The targeted design of the analogs of α-conotoxins has shown that the introduction of additional positive charges in many cases increases the affinity of α-conotoxins for both the muscle-type and neuronal nAChRs [114,115,116]. Naturally occurring α-conotoxins or their designed analogs were used to identify the respective nAChR subtypes in different biological objects—for example, on the neurons of *Lymnaea stagnalis* [117] or Turkish snail [118]—as well as to characterize the functional role of distinct receptors in these objects. The most extensive research here has been carried out on various immune cells [119,120,121,122] and cancer cell lines [123,124,125].

In collaboration with Profs. A. Smit and T. Sixma from the Netherlands, the first crystal structures of α-conotoxins were determined in complex with the AChBP from *Aplysia californica* [126,127] (Figure 3A,B). An interesting finding was that α-conotoxins, antagonists of nAChRs, induce an outward shift of AChBP loop C in the orthosteric binding site, while agonists shift it to the AChBP center. At present, this line of work is being continued with Chinese colleagues (Prof. S. Luo) by analyzing the AChBP structures in complex with conotoxins selective towards different subtypes of neuronal nAChRs [128,129,130]. In collaboration with colleagues from Greece, for the first time, the X-ray structure has been determined for the α-conotoxin RgIA, selective for the α9α10 nAChR, in complex with α9 LBD [131] (Figure 3C,D). 

The α-conotoxin RgIA analogs inhibiting the α9α10 nAChR are considered as potential drug candidates against the neuropathic pain [132,133]. In view of the important role of Arg residues in RgIA and some other “analgesic” conotoxins, at IBCh, we synthesized a series of oligoarginines (earlier used for the intracellular delivery of various attached compounds) and they were demonstrated to be a new class of inhibitors of distinct nAChR subtypes (including α9α10) [134]. In particular, octa-oligoarginine in a mouse model was shown to be as effective as α-conotoxin RgIA against oxaliplatin-induced neuropathy [135].

### 5.2. Sea Anemone-Derived Toxins

Several toxins belonging to the APETX-like family with ASIC and/or nAChR inhibitory/potentiating activity have been isolated from *H. crispa* sea anemones collected in the natural environment. Particular interest was shown in two homologous peptides, Hmg 1b-2 and Hmg 1b-4: both molecules have analgesic activity, similarly to diclofenac in the model of acid-induced muscle pain and acute local inflammation. However, Hmg 1b-4’s analgesic effect was more pronounced and statistically significant; in addition, this compound showed an anxiolytic effect on mouse mice behavior in the open field test [136]. The Hmg 1b-2 and Hmg 1b-4 (originally named Hcr 1b-4) peptides differed in their selectivity to ion channels: the second one has been found to be a potentiator of ASIC3 and inhibitor of ASIC1a at similar concentrations—EC_50_ 1.53 ± 0.07 μM and IC_50_ 1.25 ± 0.04 μM [137]. Hmg 1b-2 (originally named Hcr 1b-2) was an inhibitor of both ASIC3 with IC_50_ 15.9 μM and ASIC1a with IC_50_ 4.8 ± 0.3 μM, but at a concentration of 1 µM, it potentiated ACh-elicited currents of both α7 and α1β1δε nAChRs [137,138,139]. Then, a few novel (see Section 2) nAChR-targeting peptides from the *Metridium senile* sea anemone were discovered at IBCh and structurally and functionally characterized in collaboration with Chinese and Norwegian colleagues [21] (Figure 1 and Figure 2E).

When studying sea anemone toxins, much attention was paid to different types of receptors. Thus, inhibitors of such important ion channels as TRPV1 [140,141], TRPA1 [142,143], and ASICs [137,138,144] were discovered at IBCh. Some of the found peptides potently inhibit the mammalian nociceptive sensors, produce analgesia in animal models [136,145,146], and may pave the way to novel pain killers.

## 6. Prominent Molecules from Arthropod Venoms

Among venom-producing terrestrial animals, arthropods are unambiguous leaders both in the variety of structural folds and in the number of unique structures. First of all, there are spiders, whose venom can contain more than a hundred components. We should note that many potent and well-known molecules (in addition to the latrotoxins mentioned above) have been discovered for basic research within the walls of IBCh.

Argiopin, a non-peptide toxin from the spider *Argiope lobata*’s venom, opened the specific structural class of molecules producing a blocking effect on glutamate receptors participating in synaptic transmission [147]. Further, the number of the active molecules was extended by other compounds from *A. lobata*—argiopine, argiopinines, and pseudoargiopinines—which could block the same receptors, but their effects on the receptor channels were different from those of argiopin [148]. The latter (often called argiotoxin AgTx-636) was used with the electrophysiology and structural biology techniques to elucidate the mechanisms by which small-molecule blockers selectively inhibit ion channel conductance in calcium-permeable AMPA receptors [149] (Figure 3E). The other well-known bee venom toxin apamin, which is often cited as selective to small-conductance Ca-activated potassium channels (KCa2), was tested against 42 ion channels, KCa, KV, NaV, nAChR, ASIC, and others. As a result, its unique selectivity to KCa2 channels was confirmed [41].

In the venom of the Central Asian spider *Geolycosa sp*., the first molecule exerting powerful and selective inhibitory action on P_2_X_3_ receptors was discovered and named purotoxin-1 (PT1). Since P_2_X_3_ purinoreceptors expressed in mammalian sensory neurons play a key role in several processes, including pain perception, the peptide PT1 started a marathon of P_2_X inhibitor development with the aim of drug design. Animals treated by such peptides demonstrated anti-nociceptive behaviors in models of inflammatory pain [150]. In general, toxin use for drug development is an ongoing trend in the toxinology community. In this regard, interesting data have been obtained for the inhibition by toxins of the skeletal muscle voltage-gated sodium channel NaV_1_._4_ subtype, mutation in which is related to the congenital disease of hypokalemic periodic paralysis. Gating modifier toxins from the *Heriaeus melloteei* spider can target the mutated channel by modifying the function of the voltage sensing domain, which results in the reduction of the pathogenic gating pore currents through NaV_1_._4_ [151].

From the venom of scorpion *O. scrobiculosus*, a toxin named OSK2 has been purified and shown to be a potent and selective blocker of Kv_1_._2_ channels (K_d_ 5.97 nM) [152]. Another ligand, specific to Kv_1_._2_ channels, was purified from *B. eupeus* venom and named MeKTx11-1 (α-KTx 1.16). It had the highest affinity to KV_1.2_ (IC_50_ ∼0.2 nM), while its activity against Kv_1.1_, Kv_1.3_, and Kv_1.6_ was 10,000-, 330-, and 45,000-fold lower, respectively, as measured using the voltage-clamp technique on mammalian channels expressed in *Xenopus* oocytes [153]. Two other scorpion toxins were successfully used to obtain fluorescent tools to study potassium channels. In particular, the chimeric molecules for the toxin construction approach and full-size fluorescent protein resulted in obtaining the potassium channel blockers eGFP-OSK1 and RFP-AgTx2, which show inhibitory activity and the possibility of visualization as well [154].

For potential-activated sodium channels, a group of toxins with a principally new mechanism of action was found in the venom of spider *A. orientalis*. These toxins, consisting of 36–38 amino acid residues, were named β/δ-agatoxins and could modulate the insect NaV channel (DmNaV1/tipE) in a unique combination. They shifted the voltage dependence of the channels’ activation towards more hyperpolarized potentials (like site 4 toxins) and induced a non-inactivating persistent Na current (site 3-like action) [155].

An interesting finding was the isolation of an insect-specific toxin from *Tibellus oblongus* spider venom acting on the insect’s voltage-activated Ca channel. This 41-residue-long toxin, called ω-Tbo-IT1, produced a toxic effect with an LD_50_ of 19 μg/g on house fly *Musca domestica* larvae and with an LD_50_ of 20 μg/g on juvenile *Gromphadorhina portentosa* cockroaches. In *Periplaneta americana* cockroach neurons, one type of voltage-gated Ca^2+^ current was inhibited by ω-Tbo-IT1; thus, the toxin apparently acted as an inhibitor of presynaptic insect Ca^2+^ channels. Spatial structure analysis by NMR spectroscopy in aqueous solution revealed that the toxin comprised the modified ICK fold with an extended β-hairpin loop and short β-hairpin loop, which can make ”scissor-like” mutual motions [156].

To summarize the data about animal toxins obtained at the IBCh, we present them in the form of two tables. Table 1 gives the list of naturally occurring animal toxins discovered at IBCh, and Table 2 shows the derivatives of a number of toxins used as research tools for the study of their molecular targets.

## 7. Conclusions

In this review, we aimed to illustrate the research on peptide and protein toxins from animal venoms that started at our institute (IBCh) approximately 50 years ago. It was stimulated in part by the discovery of α-bungarotoxin and by its tremendous role both in the first characterization of nAChRs and in further research. Briefly summarized in this review is the work on animal toxins carried out at our institute in earlier years; in more detail, we present the recent achievements. The reviewed publications illustrate a very long journey from the first primary structures to the present-day role of peptide and protein neurotoxins in research on nAChRs and on different types of ion channels. Apart from neurotoxins, diverse toxins acting on various biological targets have been studied as well. We also hope that the ongoing research is presented in sufficient detail. It should be emphasized that many excellent labs in the world were and are working on protein and peptide neurotoxins, and the fruitful collaboration of IBCh with them is mentioned in the text and is reflected in the author names in the References.

## Figures and Tables

**Figure 1 ijms-24-13884-f001:**
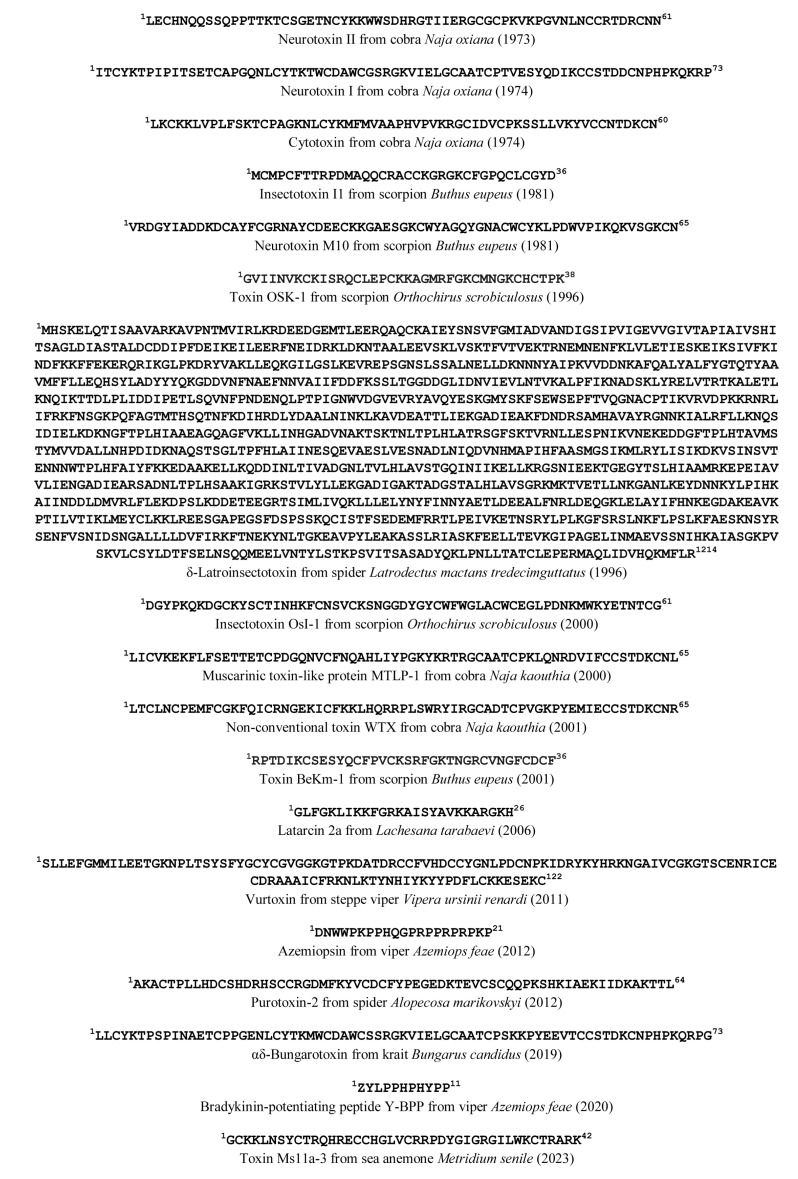
Selected amino acid sequences of animal toxins discovered at IBCh.

**Figure 2 ijms-24-13884-f002:**
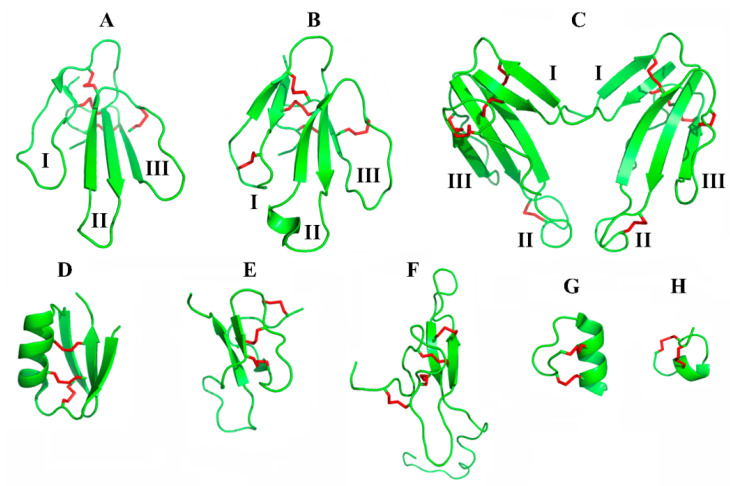
NMR structures of selected representatives of animal toxins obtained in different years at IBCh. (**A**) Short-type neurotoxin NTII from cobra *N. oxiana* (PDB ID: 2MJ4); (**B**) [P33A]-mutant of non-conventional toxin WTX from cobra *N. kaouthia* (2MJ0); (**C**) dimeric form of long-type α-cobratoxin from cobra *N. kaouthia* (4AEA); (**D**) OSK1 from scorpion *O. scrobiculosus* (1SCO); (**E**) Ms11a-3 from sea anemone *M. senile* (6XYI); (**F**) purotoxin-2 from spider *A. marikovskyi* (2MZF); (**G**) apamin from bee *A. mellifera* (7OXF); (**H**) α-conotoxin ImI from *C. imperialis* (1IMI). Disulfides in all toxins are shown in red. Roman numerals indicate loop numbers in TFTs: I—N-terminal loop, II—central loop, III—C-terminal loop.

**Figure 3 ijms-24-13884-f003:**
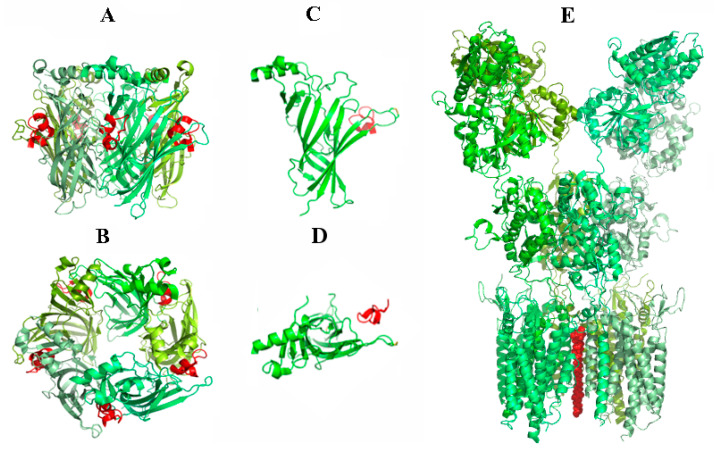
Three-dimensional structures of toxin–receptor complexes in the determination of which scientists from IBCh took part. (**A**,**B**)—side and top views, respectively, of X-ray structure of *Aplysia californica* AChBP in complex with α-conotoxin PnIA[L10,K14] (PDB ID: 2BR8); (**C**,**D**)—side and top views, respectively, of X-ray structure of extracellular domain of human α9 nAChR in complex with α-conotoxin RgIA (6HY7); (**E**)—side view of cryo-electron structure of calcium-permeable constructed AMPA receptor (GluA2_Q_-STZ) in complex with argiopin, also known as argiotoxin-636 (AgTx-636) (6O9G). All toxins are shown in red. Identical five AChBP protomers as well as four AMPA receptor subunits are colored in different variants of green for clarity.

**Table 1 ijms-24-13884-t001:** Some representatives of animal toxins discovered and investigated at IBCh.

Toxin(s)	Species	Structure Type	Structural and Functional Characterization: Sequence/Spatial Structure/Functional Characterization	UniProt Code	Year of Discovery	Ref.
**Snake venom-derived toxins**						
Neurotoxin II	*Naja oxiana*	TFT: Short-type α-neurotoxin	+/+/+	3S11_NAJOX	1973	[3,31,34,36,69]
Neurotoxin I	*N. oxiana*	TFT: Long-type α-neurotoxin	+/−/+	3L21_NAJOX	1974	[4,69,71]
Cytotoxins	*N. oxiana, N. kaouthia*	TFT	+/+/+	3SA1(2)_NAJOX, 3SA3(7A,8)_NAJKA	1974–2022	[5,6,9,38,74,75,76,77,78,79,80,81,82,83,157]
Muscarinic toxin-like protein 1	*N. kaouthia, N. melanoleuca*	TFT	+/−/+	3SUC1_NAJKA, 3SUC1_NAJME	2000–2021	[8,71]
WTX	*N. kaouthia*	Non-conventional TFT	+/+/+	3NO2_NAJKA	2001	[7,35,37,69,84,85,86]
Homodimer of α-cobratoxin	*N. kaouthia*	TFT: Dimer of long-type α-neurotoxin	+/+/+	3L21_NAJKA	2008	[38,39]
Heterodimeric neurotoxic phospholipase A2	*Vipera nikolskii*	Secreted PLA2 of group II	+/−/+	PA2B1(2)_VIPBN PA2H_VIPBN	2008	[10]
Phospholipases A2	*V. ursinii renardi, Bungarus fasciatus, Bitis arietans,* *N. oxiana, N. haje*	Secreted PLA2 of group I and II	+/−/+	PA2A2(A3,B)_VIPRE PA2B_BITAR PA2B6_BUNFA PA2TI_NAJHH	1977–2011	[16,93,94]
Azemiopsin	*Azemiops feae*	Linear peptide	+/−/+	AON_AZEFE	2012	[11,101]
αδ-Bungarotoxin (αδ-BgTx-1)	*Bungarus candidus*	TFT: Long-type α-neurotoxin	+/−/+	3L21_BUNCA	2019	[72]
Bradykinin- potentiating peptides	*A. feae*	Linear peptide	+/−/+	No code	2020	[12]
Tx-NM2, Tx-NM3-1	*N. melanoleuca*	TFT: Long-type α-neurotoxin	+/−/+	3L23(24)_NAJME	2021	[71]
**Scorpion venom-derived toxins**						
Neurotoxin BeKm-1	*Mesobuthus eupeus*	α-helix and a triple-stranded antiparallel β-sheet	+/+/+	KGX21_MESEU	1996	[45,158]
**Spider venom-derived toxins**						
β/δ-Agatoxin-5	*Agelena orientalis*	ICK	+/−/+	T5G1A_AGEOR	2010	[155]
Argiopin, argiopinins, and pseudoargiopinins	*Argiope lobata*	Acylpolyamines	+/+/+	CID: 122294 189479 189486 *	1989	[159]
Latarcin-1	*Lachesana tarabaevi*	Linear peptide	+/+/+	LAT1_LACTA	2006	[102,160]
Latarcin-2a	*L. tarabaevi*	Linear peptide	+/+/+	LAT2A_LACTA	2006	[102,161]
Purotoxin-1 (PT1)	*Alopecosa marikovskyi*	ICK	+/+/+	TXPR1_ALOMR	2010	[150]
α-Latrototoxin-Lt1a	*Latrodectus mactans tredecimguttatus*	Multidomain organization	+/−/+	LATA_LATTR	1990	[27]
α-Latroinsectotoxin- Lt1a	*L. mactans tredecimguttatus*	Multidomain organization	+/−/+	LITA_LATTR	1993	[28]
**Sea anemone venom-derived toxins**						
Analgesic polypeptide HC1 (τPI-SHTX-Hcr2b)	*Heteractis crispa*	Kunitz-type	+/−/+	VKT2B_HETCR	2008	[140]
π-AnmTX Ugr 9a-1	*Urticina grebelnyi*	β-hairpin structure	+/+/+	TX9A_URTGR	2013	[144]
τ-AnmTX Ms 9a-1	*Metridium senile*	β-hairpin structure	+/−/+	TX91O_METSE	2017	[142]
Ms11a-1/4	*M. senile*	ICK	+/+/+	No code	2023	[21]

* These are references to the structures in PubChem. The toxins are of non-peptide nature.

**Table 2 ijms-24-13884-t002:** Animal toxins and their derivatives developed as tools in IBCh.

Native Toxin (Species)	Modifications	Target(s)/Activity	Tasks	Ref.
α-Bungarotoxin (*Bungarus multicinctus*)	Radiolabeled and fluorescent derivatives	αβδγ/ε-, α7-, α9(α10) nAChRs; AChBPs; GABA-A	Detection of respective targets in different preparations, cells, tissues Radioligand in radioligand assay	[7,11,35,38,39,69,71,73]
Other long-type α-neurotoxins	Radiolabeled, photoactivatable, and fluorescent derivatives, analogs	αβδγ/ε-, α7-, α9(α10) nAChRs; AChBPs; GABA-A	Detection of respective targets in different preparations, cells, tissues Radioligand in radioligand assay Structure–function characterization Mapping of ligand-binding sites of *T. californica* nAChR	[4,34,68,69,71,73]
κ-Bungarotoxin (*Bungarus multicinctus*)		α7-, α3β2 nAChRs	Detection of respective targets in different preparations	[38]
Short-type α-neurotoxins	Radiolabeled and photoactivatable derivatives, analogs	αβδγ/ε nAChRs	Detection of respective targets in different preparations Determination of the spatial structure Structure–function characterization Mapping of ligand-binding sites of *T. californica* nAChR	[3,34,65,66,67,69]
Phospholipases A2		Cytotoxicity, antiviral activity, thrombin inhibitor, αβδγ/ε-, α7-nAChRs	Detection of respective targets in different preparations, cells, tissues Revealing the different biological activity	[95,96,97,98,99,100]
Cytotoxins	Fluorescent derivatives	Cytotoxicity	Determination of the spatial structure Determination of the cytotoxicity mechanism	[138]
Azemiopsin (*Azemiops feae*)	Ala analogs, fluorescent derivative	αβδγ/ε nAChRs	Binding and functional characterization Structure–function characterization Preclinical trials as myorelaxant	[11,101]
Agitoxin-2 and OSK-1	Toxin fused with fluorescent proteins	Voltage-gated potassium channels (KV)	Ligand screening in the spheroplast binding assay	[154]
Agitoxin-2	Fluorescently labeled derivatives	Spheroplasts with the embedded KcsA-Kv1.3 hybrid protein	Screening of prospective compounds recognized by Kv1.3	[162]
Toxin PT1	Pure pharmacological substance	P_2_X_3_	Preclinical trials as analgesic	-
APHC3	Pure pharmacological substance	TRPV1	Preclinical trials as analgesic	-
Different α-conotoxins (*Conidae*)	Different analogs, radiolabeled, photoactivatable, and fluorescent derivatives	Different nAChR subtypes; AChBPs; GABA-A	Determination of the spatial structure Detection of respective targets in different preparations, cells, tissues Radioligand in radioligand assay Design of new analogs Mapping of ligand-binding sites of *T. californica* nAChR X-ray studies of AChPB complexes In vivo studies on neuropathy models	[40,69,72,108,109,110,111,112,113,114,115,116,117,118,119,120,121,122,123,124,125,126,127,128,129,130,131,135]
αO-Conotoxin GXIVA (*Conus geographus*)	Isomers, radiolabeled derivative	α9(α10) nAChR; AChBPs	Binding and functional characterization	[163]

## Data Availability

Not applicable.

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
