# Peer review of "Fifty Years of Animal Toxin Research at the Shemyakin–Ovchinnikov Institute of Bioorganic Chemistry RAS"

_ijms, 2023, doi:10.3390/ijms241813884_

Round 1

Reviewer 1 Report

This is a comprehensive review of the work carried out at the Shemyakin Ovchinnikov Institute of Bioorganic Chemistry on their pioneering work on animal toxins, including α-bungarotoxin-like three-finger toxins, latrotoxins, snake-derived neurotoxins, and cone snail toxins, acting at nAChR and other ion channel targets. Much of this work has been highly influential in the neurotoxins field, and this review is a useful summary and tribute to the contributions of the IBCh. 

I recommend publication, but would ask the authors to consider amending the comment on Line 528: "We gave references to some of old publications from IBCh which apparently are not very interesting to a present-day researcher." 

Author Response

Dear Reviewer, thank you for your kind report, we have changed the sentence at Line 528 as you suggested. The new version is "Briefly summarized in this review the work on animal toxins done at our institute in earlier years, in more detail we are presenting the recent achievements".

Reviewer 2 Report

This report describes the main achievements of the last 50 years of work on animal toxins at the Shemyakin-Ovchinnikov Institute of Bioorganic Chemistry (IBCh) of the Russian Academy of Sciences. During all these years, a lively and fruitful collaboration has been established with related laboratories and groups outside Russia. Remarkable and in many cases groundbreaking results have been published in numerous articles.

The main object of research is proteins and peptides from animal venoms, including snakes, scorpions, spiders and marine animals, which act on nicotinic acetylcholine receptors (nAChRs) and other ion channels. Most toxins have been thoroughly characterised structurally and functionally, including determination of their primary structures by Edman and cDNA sequencing, mass spectrometry and transcriptome analyses, spatial structures by 1H-NMR and detailed mechanisms of action by various electrophysiological, molecular modelling, cell-based and spectral imaging techniques. Several types of toxins have been characterised from snake venoms, e.g. three-finger toxins and phospholipases A2. The discovery of a new toxin from Azemiops feae, a linear peptide called azemiopsin, which selectively inhibits the muscle-type of nAChRs, established a large family of polypeptides with similar structural motifs. In preclinical studies, azemiopsin acted as an efficient myorelaxant.

At the IBCh, intensive research is being conducted on marine toxins from marine snails of the genus Conus and sea anemones. In addition to the work on characterising the neurotoxic effect of alpha-conotoxins, their effect on various immune cells and cancer cell lines has also been studied. Of particular interest are the toxins derived from sea anemone of different types, Kunitz-type, beta-hairpin structure and ICK, which were able to inhibit the ion channels TRPV1, TRPA1 and ASICs. Some of them produce analgesia in animal models and could pave the way to novel analgesics. Last but not least, very interesting toxins have been discovered in arthropod venoms, especially spider and scorpion venoms, many of which show unique selectivity towards specific ion channels, making them promising candidates for drug development to treat chronic pain or as research tools to study their receptors.

Indeed, many of the above animal toxins and their radiolabelled, fluorescently labelled and photoactivatable derivatives have been developed at the IBCh to perform various tasks in structure/function/interaction studies.

Only one minor typo was found in the manuscript at line 387, where "5.1.a-C. onotoxins" should be corrected. There are no further comments on the manuscript.

Author Response

Dear Reviewer, thank you for your kind report, we have corrected the typo at Line 387 as you suggested. The new version is "5.1.α-Сonotoxins in research on nAChRs".